

# Sea Ice Model Intercomparison Project (SIMIP): Understanding sea ice through climate-model simulations

Dirk Notz[1], Alexandra Jahn[2], Marika Holland[3], Elizabeth Hunke[4], François Massonnet[5,6], Julienne Stroeve[7,8], Bruno Tremblay[9], and Martin Vancoppenolle[10]

[1]Max Planck Institute for Meteorology, Hamburg, Germany
[2]Department of Atmospheric and Oceanic Sciences and Institute of Arctic and Alpine Research, University of Colorado at Boulder, Boulder, USA
[3]Climate and Global Dynamics Laboratory, National Center for Atmospheric Research, Boulder, USA
[4]Theoretical Division, Los Alamos National Laboratory, Los Alamos, New Mexico, USA
[5]Earth Sciences Department, Barcelona Supercomputing Center (BSC-CNS), Barcelona, Spain
[6]Georges Lemaître Centre for Earth and Climate Research, Earth and Life Institute, Université catholique de Louvain, Louvain-la-Neuve, Belgium
[7]National Snow and Ice Data Center, Boulder, USA
[8]University College London, London, UK
[9]Department of Atmospheric and Oceanic Sciences, McGill University, Montreal, Canada
[10]Sorbonne Universités, UPMC Paris 6, LOCEAN-IPSL, CNRS/IRD/MNHN, France

*Correspondence to:* D. Notz (dirk.notz@mpimet.mpg.de)

**Abstract.** A better understanding of the role of sea ice for the changing climate of our planet is the central aim of the diagnostic CMIP6 Sea-Ice Model Intercomparison Project (SIMIP). To reach this aim, SIMIP requests sea-ice related variables from climate-model simulations that allow for a better understanding, and ultimately improvement, of biases and errors in sea-ice simulations with large-scale climate models. This then allows us to better understand to what degree CMIP6 model simulations relate to reality, thus improving our confidence in answering sea-ice related questions based on these simulations. Furthermore, the SIMIP protocol provides a standard for sea-ice model output that will streamline and hence simplify the analysis of the simulated sea-ice evolution in research projects independent of CMIP. To reach its aims, SIMIP provides a structured list of model output that allows an examination of the three main budgets that govern the evolution of sea ice, namely the heat budget, the momentum budget and the mass budget. In this contribution, we explain the aims of SIMIP in more detail and outline how its design allows us to answer some of the most pressing questions that sea ice still poses to the international climate-research community.

## 1 Introduction

Sea ice is both a key indicator and a driver of climatic changes on our planet. In addition, the temporal and spatial evolution of its coverage has important implications far beyond climatic changes: polar marine biogeochemistry and ecosystems are closely related to the existence of sea ice (e.g., Kovacs et al., 2010; Vancoppenolle et al., 2013; Tynan, 2015), as is the livelihood of indigenous populations at the shores of the Arctic Ocean (e.g., Laidler et al., 2008). Reductions in sea ice are allowing for





increased tourism (e.g., Stewart et al., 2010), regular shipping through the Northern Sea Route (e.g., Liu and Kronbak, 2010; Smith and Stephenson, 2013; Stephenson et al., 2014) and resource extraction. In addition, sea ice is a crucial component of the prevailing cultural view of high latitudes as a frozen landscape, which is also reflected by the usually wide-spread media and public attention that prospects of a seasonal sea-ice free Arctic obtain.

In light of this importance of sea ice from a wide range of stakeholders, it is sobering to see how much simulations of its past and future evolution differ across large-scale coupled models (e.g., Massonnet et al., 2012; Stroeve et al., 2012), how much retrieved sea-ice properties from one satellite product differ from another satellite product (e.g., Meier and Notz, 2010; Ivanova et al., 2015) , and in how many aspects the simulations and observations differ from each other (e.g., Massonnet et al., 2012; Stroeve et al., 2012; Turner et al., 2013; Stroeve et al., 2014; Gagné et al., 2015; Shu et al., 2015). We do not yet know how

much these differences are irreducible, for example because they are due to internal variability of the climate system, and how much they reflect biases in the model's representation of the functioning of the climate system in high latitudes. This lack of understanding hinders further improvements of our models, an identification of observational needs, and a robust assessment of the most likely future evolution of sea ice in response to the ongoing climatic changes on Earth.

   To address these issues, a specific model intercomparison project related to sea ice, namely SIMIP, has been endorsed

as an official part of the 6th phase of the Coupled Model Intercomparison Project (CMIP6, Eyring et al., 2015), which is coordinated by the World Climate Research Programme (WCRP). Among the 21 endorsed model intercomparison projects (MIPs), SIMIP is one of four so-called diagnostic MIPs. Hence, in the interest of the most efficient use of the resources of the climate-research community, SIMIP does not request any dedicated experiments but instead provides the infrastructure to analyze experiments carried out by other MIPs from a sea-ice perspective by requesting sea-ice output. This then allows us to

address both challenging sea-ice related research questions that are of interest for a wide range of scientists and stakeholders, and to address questions that are of immediate interest for sea-ice scientists.

   This contribution describes the aim and design of SIMIP in detail. In particular, we describe the requested sea-ice related variables that are included in the CMIP6 data call. Our hope is that in doing so, we provide a framework that allows for the detailed, unified, and efficient analysis of sea-ice simulations well beyond CMIP6.

We first outline briefly some of the most pressing research questions that we hope to address through SIMIP and place them within the CMIP6 context. We then move on to discuss in more detail the philosophy behind the design of SIMIP, which then will hopefully also allow for a consistent future extension of the SIMIP protocol as sea-ice models become more and more sophisticated. In an appendix, we specify and define in detail the current list of requested sea-ice related variables from CMIP6 simulations, giving concrete guidance on the variable definitions to model developers and modeling centers that would like to store these variables from their simulations. The diagnostic sea-ice output should be saved as much as possible in all CMIP6 experiments that include a sea-ice-model component.



## 2   Guiding questions of SIMIP

The overarching aim of SIMIP is very simple. We want to understand how sea ice works and evolves in the coupled climate
system of our planet. As discussed by Stroeve and Notz (2015), such understanding is only possible through combining model
simulations with observations. This is because model simulations allow us to quantify and understand the interaction of ice,
air and ocean in the modeling domain, while the observational record then allows us to infer if the model simulations capture
the behaviour of the real world. Hence, within SIMIP we aim both at an understanding of the behaviour of sea ice in the model
world, but also at an understanding of the degree to which this model world describes the real world, which then allows us to
identify ways by which this agreement can be improved. Ultimately this will allow for more realistic simulations of the sea-ice
cover, including more robust projections of its future evolution.

Reflecting this line of thinking, we tend to differentiate the guiding questions of SIMIP into three distinct sets: First, why do
model simulations differ from each other? Second, why do model simulations differ from the observational record? And third,
what can we do to reduce these differences to obtain a better understanding of sea ice in the climate system and eventually
to achieve more realistic projections of the sea-ice evolution in both hemispheres? These guiding questions of SIMIP address
aspects of all three science questions of CMIP6 (Eyring et al., 2015), with a particular focus on identifying systematic biases
in sea-ice simulations and on understanding the response of sea ice to forcing. Overall, by addressing these guiding ques-
tions, SIMIP will advance the understanding of sea ice within the climate system, which aligns with the goals of the WCRP
Cryosphere Grand Challenge.

In answering the three questions, our analysis is based on the understanding that model simulations are never able to reflect
reality, but that they can be close enough to a reflection of reality that they become useful. As discussed by Notz (2015), such
reflection of usefulness is not readily obtained from a mere agreement of model simulations with observations, as is often
assumed. Instead, the usefulness of a climate model simulation can only be confirmed if, among others, the internal variability
of the variable of interest, the uncertainty of the observational record, and the tuning of the models are also considered.
These issues are reflected by the scientific plan that underlies SIMIP, as outlined in the following description of our guiding
questions. Note that we group these questions according to the three sets just described, whereas in practice often several of
these guiding questions will be answered simultaneously. Note also that the guiding questions that we outline here only reflect
our understanding of some key open research topics that are on our agenda today. The design of SIMIP is, as outlined in the
following section, intentionally generic enough to also allow us to answer those sea-ice related questions that will only emerge
after the simulations for CMIP6 have been carried out.

### 2.1   Understanding differences between model simulations

The disagreement among state-of-the-art climate model simulations regarding, for example, the past and future evolution of
Arctic and Antarctic sea ice is striking, and has in particular not reduced much from CMIP3 to CMIP5 (Stroeve et al., 2012;
Turner et al., 2013). This large spread casts severe doubt on our ability to robustly project the long-term evolution of parts of the
Earth's climate system with today's climate models. It is, however, currently not clear to which degree both the model spread



and the disagreement of modeled large-scale sea-ice evolution from the observational record is a reflection of the insufficient quality of the sea-ice component of state-of-the-art climate models. This is because such disagreement can be the result of a combination of a large number of underlying causes. These include internal variability, different approaches for tuning the models, the forcing of the simulated sea ice from the atmosphere and the ocean, and shortcomings of individual sea-ice model

formulations. To further complicate matters, the mix of answers might be different for the Arctic and Antarctic sea ice. The influence of so many different factors shows that it is not sufficient to only quantify differences between model simulations, but that we must understand the underlying cause of these differences in order to meaningfully make progress in the quality of sea-ice simulations and in our understanding of sea ice within the climate system.

To obtain such understanding is one of the central aims of SIMIP, following the spirit of a previous SIMIP exercise carried

out about two decades ago (Lemke et al., 1997). A particular focus is placed on internal variability, which has been shown to explain much of the model spread of simulated Arctic sea-ice trends (e.g., Swart et al., 2015; Notz, 2015) and at least some of the spread between Antarctic sea-ice trends (Mahlstein et al., 2013; Gagné et al., 2015). While a rough analysis of the quantitative model spread and of internal variability has already been possible with the limited set of sea-ice related variables stored in CMIP5, the SIMIP data request will allow us to more specifically understand differences in the drivers of internal

variability across different models. This is because the new variables requested by SIMIP allow us to better differentiate between atmosphere versus ocean driven variations in sea-ice coverage, as well as thermodynamic versus dynamic variations. The variables that are specifically useful for this purpose relate to the individual atmospheric fluxes over the sea-ice covered part of any grid cell that SIMIP requests, rather than the average fluxes over the entire grid cell including the ice-free part that used to be requested in previous CMIPs. This is because the latter does not allow one to analyse the heat budget of the ice in

detail. Regarding the temporal evolution, compared to CMIP5 three additional variables (snow thickness, surface temperature of sea ice, sea-ice speed) are requested as daily averages, in addition to the standard monthly averages. Among other research topics, this higher temporal resolution allows an in-depth assessment of sea-ice-melt onset and freeze-up variability in the different models.

The tuning of sea-ice models (or lack thereof) is another factor that might explain the spread of the simulated sea-ice

evolution in CMIP5 model simulations. As part of the SIMIP protocol, we hence request documentation of the tuning procedure of the models (see Appendix A). This knowledge will allow us to evaluate the potential impact of model tuning on the sea-ice metrics from different models (see, for example, Gough, 2001; Mauritsen et al., 2012; Notz et al., 2013). Furthermore, based on this information from the different modeling groups, it might even be possible to establish a guide to best practices regarding the tuning of sea-ice models in coupled climate models.

In addition to internal variability and tuning, the forcing of the atmosphere and ocean influence the evolution of sea ice in model simulations, independent of the specific quality of the sea-ice model component. For example, in the Antarctic, the influence of biases in the mean state and trends of the atmosphere (Mahlstein et al., 2013; Haumann et al., 2014; Purich et al., 2016) and ocean (e.g., Armour and Bitz, 2015) simulations are thought to be the main cause for biases in the simulated sea-ice

cover. SIMIP will allow researchers to assess in more detail the influence of the atmospheric and oceanic drivers on the sea-ice cover, thus allowing one to quantify the role of oceanic and atmospheric biases for biases in the simulated sea-ice evolution.





In combining our understanding of internal variability, the tuning of individual models, and the influence of atmospheric and oceanic mean state biases on the sea-ice simulation, we will be able to assess how much these factors can explain differences, both in the control simulation of individual models and across CMIP6 experiments, in particular the CMIP6 historical (Eyring

et al., 2015) and ScenarioMIP (O'Neill et al., 2016) experiments. Due to the different characteristics and mean states in the two polar regions, the answers we obtain might be entirely different for Arctic and Antarctic sea-ice simulations, but will in any case allow us to identify the main aspects in which large-scale climate models need to be improved to obtain better simulations. This knowledge will then enable us to provide guidance for the design of large-scale experimental programs such as the upcoming Year of Polar Prediction (2017–2019, www.polarprediction.net/yopp) or the MOSAiC campaigns (2019–2021,

www.mosaicobservatory.org) that aim to perform specific measurements useful for sea-ice-model development.

## 2.2   Understanding differences between model simulations and observations

In CMIP5, model simulations for both Antarctic and Arctic sea-ice extent differ from observations (e.g., Massonnet et al., 2012; Stroeve et al., 2012; Turner et al., 2013; Stroeve et al., 2014; Gagné et al., 2015; Shu et al., 2015), but it is unclear what the underlying cause of these differences are. Based on an improved understanding of inter-model differences in sea-ice

simulations and an assessment of the magnitude of internal variability in the simulations, we will assess the differences between CMIP6 model simulations and the observational records. In the past, such differences were often only pointed out and then directly taken as proof for shortcomings of individual models. Within SIMIP, however, we aim to understand the differences between models and observations. A central question, in particular, is whether the disagreement might simply be a reflection of internal variability rather than be caused by a model shortcoming, as described in section 2.1. If it turns out that the difference

reflects a true inconsistency between model simulation and reality, we will then examine causes for the identified biases within SIMIP. We will do so on the working premise that such inconsistencies are either a reflection of observational shortcomings, of shortcomings in the models or, as most likely will often be the case, a combination of both.

To robustly identify model shortcomings, we will use a plausibility variable as described by Santer et al. (2008) and Stroeve et al. (2012) that considers both internal variability and observational uncertainty. While internal variability will be estimated

directly from the model simulations, for example using the method of Hawkins and Sutton (2009) or insights from large ensemble simulations with an individual model (Swart et al., 2015), observational uncertainty will be obtained through close cooperation with the satellite community. When the difference between model simulation and observations is larger than the combined influence of observational uncertainty and internal variability, we have robustly identified a shortcoming of a specific model simulation. We will then apply the same methods that we described for the analysis of inter-model differences to establish the underlying reason for the model bias, and, in particular, identify whether biases are caused by issues with the sea-ice models or rather by issues in the oceanic or atmospheric forcing. Such assessment will be done through a cross-model assessment of the underlying drivers of specific biases, which will be possible on a process level by the variables that we define through the SIMIP protocol. For example, if we find models to simulate biases in surface albedo in summer as shown by Koenigk et al.

(2014), we can use the new additional variables for melt-pond coverage, snow coverage and sea-ice concentration to identify



the main reason for a specific model's failure to capture the observed evolution of albedo. This can then be used to improve the model simulations, which in turn allows us to better understand the role of sea ice for the changing climate of our planet.

## 2.3 Understanding and predicting sea ice

Sea-ice models, as well as all other components of Earth System Models, are usually not developed as an end in themselves.

Instead, they are developed to answer specific research questions, which arise, for example, out of the curiosity of a scientist, out of the aim to most robustly increase profits in economic ventures, or out of a societal interest to allow for the development of the best future policies. Independent of the motivation of any such questions, the robustness of our answers hinges on the faith we have in our models to realistically represent the main processes relevant for a specific question. As such, the understanding of model biases or the quantification of internal variability are "just" prerequisites to then eventually give answers to the truly

relevant research questions that we aim to answer.

Hence, while we couldn't answer these questions without the prerequisites just described, the success of SIMIP will eventually be measured against the degree to which it allows us to answer a wide variety of research questions. Many such questions will be posed and answered by scientists not directly involved in SIMIP work, but of course also within SIMIP we aim to provide answers from CMIP6 model simulations that are not possible without the detailed understanding of sea ice that the SIMIP

protocol allows. These aims may sometimes simply be achieved by allowing us to more robustly identify model simulations that are more trustworthy than others, or by allowing a more robust understanding of individual processes which can then guide additional research in narrowing down uncertainty for some of the most widely discussed questions related to sea ice.

For example, we currently have a limited understanding of the potential or real predictability of the evolution of sea ice on time scales from weeks to decades. In recent years, several initiatives have formed to address the short-term predictability

of sea ice (in particular, the Sea ice Prediction Network (SIPN), Polar Climate Prediction Initiative (PCPI), and the Polar Prediction Project (PPP)). A dedicated international research program focused on polar prediction is coordinated through the Year of Polar Prediction (YOPP, Goessling et al. (2015)), and we expect model simulations following the SIMIP protocol to provide a wealth of useful data that will be analysed through YOPP. In particular, we expect that the improved understanding of processes that SIMIP makes possible will allow us to more robustly understand the limits of sea-ice predictions.

We currently also have very little understanding of the long-term evolution of sea ice in both hemispheres. For instance, CMIP5 RCP8.5 simulations show a spread in the timing of seasonal ice-free conditions in the Arctic from 2005 to well beyond 2100 (e.g., Massonnet et al., 2012). This spread reflects the different levels of climate sensitivity of models as well as the different levels of the sensitivity of sea ice to global warming. This in turn severely limits our ability to robustly answer more fundamental questions of the role of sea ice in the Earth's climate system. Hence, reducing the spread of sea-ice projections is not an aim in itself, but doing so based on an understanding of the root causes of the spread will also increase our fidelity in our answers to other sea-ice related questions in climate research.





## 3   Design of SIMIP

Our guiding philosophy during the design of SIMIP is to most efficiently prepare today for the sea-ice related questions we
might be asking tomorrow. This philosophy was born out of our shared frustration that many sea-ice related questions that
came up during CMIP5 or other similar exercises simply could not be answered because the necessary model output was not
available. Hence in designing SIMIP, we compiled a consistent list of sea-ice related variables that allow us to understand the
main physical drivers that govern the evolution of sea ice. These are described by the conservation of heat, the conservation
of momentum, and the conservation of mass. To make the description of the required sea-ice variables as easy to follow as
possible, we divided the output variables into five groups, each describing a key aspect of sea-ice evolution. These groups
are explained in more detail in this section. They are (1) sea-ice state variables, (2) tendencies of sea-ice mass, (3) heat and
freshwater fluxes, (4) sea-ice dynamics and (5) integrated quantities.

For each variable, we specify a priority that describes how crucial knowledge of this variable is for our understanding of sea
ice. As a guiding principle in defining the priorities, we roughly grouped variables relative to the number of researchers most
likely to use a given variable.

Priority 1 variables are those variables needed to quantify the large-scale evolution of sea ice or to understand the forcing of
sea ice on either the ocean or the atmosphere. These variables will be used by many scientists, even those without a specific
interest in sea ice *per se*, and should be stored as core sea-ice variables in any large-scale model simulation.

Priority 2 variables are those needed to understand the detailed evolution of sea ice in response to external forcing. These
variables will be used by researchers who want to understand in more detail which formulations of a model drive the bulk
behaviour of the modeled sea ice. All priority 2 variables should readily be available from most modern sea-ice models.

Priority 3 variables are variables that are primarily helpful for scientists who develop sea-ice models, assess the detailed
sea-ice related budget, or carry out detailed comparisons of simulation results with field observations. These variables often
are not used by the models in their standard calculation, so they need to be calculated specifically as diagnostics to understand
the model behavior. Also variables that are not available from some models because they do not (yet) include a specific process
usually fall into priority 3.

### 3.1   Sea-ice state variables

The variables that we label as "sea-ice state variables" describe the large-scale state of sea ice. They are made up of the
distribution of sea-ice mass and variables that allow us to assess the total heat content stored within the ice.
To assess sea-ice mass, the SIMIP protocol requests priority 1 variables such as sea-ice concentration in individual grid
cells, sea-ice thickness, sea-ice mass per grid-cell area and variables related to the snow coverage on sea ice. At priority 3,
additional variables are requested that describe more detailed properties of the simulated sea-ice cover, for example related to
the sub-grid-scale distribution of ice thickness, the amount of sea ice in ridges or the distribution of melt ponds.

The main variable describing the amount of sea ice in large-scale model simulations used to be sea-ice volume, which usually
was provided as "sea-ice volume per grid area". Since this variable had the units "m", it was often referred to as "equivalent





thickness", which caused confusion if researchers thought that this variable described actual thickness. However, the actual conserved quantity is sea-ice mass, which is why we have decided to prefer its storage over that of volume. In addition, we have dropped the misleading notion of "equivalent thickness", and directly request "sea-ice volume per grid area". This variable is partly kept for consistency with earlier CMIPs, and partly because the underlying concept of equivalent thickness remains possibly useful from an ocean perspective.

In addition to sea-ice mass, the SIMIP protocol requests the actual thickness of the simulated sea ice, averaged over the ice-covered part of the grid cell. From a sea-ice perspective, this actual thickness is geophysically more meaningful than the previous notion of equivalent thickness, as the properties of the ice cover usually depend more directly on actual thickness than on the synthetic equivalent thickness. In particular, we hope that this new definition is more intuitively meaningful and thus avoids misinterpretation of sea-ice thickness by researchers unfamiliar with the concept of equivalent sea-ice thickness.

To assess the thermal properties of sea ice, the SIMIP protocol requests at highest priority those variables that most directly affect the atmosphere above the ice and the heat exchange between the ice and the ocean. Hence, observables such as sea-ice albedo or surface temperature are requested at priority 1. At lower priority 2, we request variables that describe in more detail the thermal state of the sea-ice cover, such as the temperature at the snow–ice interface, formally defined as "sea-ice surface temperature" by the NetCDF Climate and Forecast (CF) protocol.

Because of the central role of snow and sea-ice thickness, areal concentration and surface temperature in describing the sea-ice evolution in coupled models, these are among the few variables that SIMIP requests at daily resolution. In contrast to CMIP5, we have added the amount of snow to the list of daily variables, since it has previously not been possible to examine the onset of surface melt at a temporal resolution of less than a month, which severely limited the usefulness of respective observations for model evaluation.

## 3.2   Tendencies of sea-ice mass

To understand the change of sea-ice mass in different models, SIMIP requests variables quantifying the physical cause and location of ice growth and melt. They are all requested at priority 2, and capture both the areal and the thickness evolution of the ice cover. Among others, the evolution of ice thickness through lateral growth or melt, through bottom growth or melt, through surface melt, snow–ice formation etc. are requested as part of this variable group. Details of the processes causing

changes in snow mass are included also, i.e. advection, snowfall, snow melt etc. These variables then allow one to identify at first order the physical processes that change the sea-ice and snow mass in different models. This then in turn allows one to more robustly identify the underlying processes that need to be better represented by individual models to better capture the observed evolution of the ice cover.

## 3.3   Heat and freshwater fluxes

Only quantifying where the sea-ice mass is changing does not allow us to understand why the sea-ice mass is changing the way it does. For such analysis, atmospheric and oceanic heat fluxes that affect the sea-ice cover are required, and SIMIP requests these fluxes at priority 2. In the past, individual atmospheric fluxes over sea ice have not been available, since usually only their





grid-cell average, including the ice-free part of the grid cell, was recorded. This then made it difficult to assess how the fluxes developed over the sea-ice covered part of the grid cell, since a change in the net fluxes could also simply have been a reflection of changing sea-ice concentration. Therefore within SIMIP, we specifically request all fluxes over the ice-covered part of the grid cell. Individual atmospheric fluxes are often only available on the atmospheric grid, with the sea-ice model only receiving an averaged bulk flux interpolated to the oceanic grid. This averaged flux then does not allow for a detailed analysis of the heat

budget. Since we do not specify on which grid the individual fluxes should be provided, modelling centres are free to simply provide the individual atmospheric fluxes on the atmospheric grid where they are readily available in any case.

In addition to the heat fluxes, an analysis of sea-ice mass changes also must take relevant freshwater fluxes into account, e.g. rainfall or freshwater flux at the bottom of the ice from sea-ice phase changes. To examine the salt release from sea ice and its possible impact on ocean circulation, SIMIP also requests storage of the sea ice related salt flux.

## 15   3.4   Sea-ice dynamics

While the variables requested so far allow researchers to analyze in great detail the change of sea-ice mass in different models and in any given grid cell, the movement of sea ice is also of central importance to understand the evolution of Arctic sea ice as simulated by large-scale climate models. Hence, SIMIP requests variables that describe this movement and that allow researchers to understand its driving forces in CMIP6 simulations.

At the highest priority 1, the SIMIP data request includes for sea-ice dynamics the $x-$ and $y-$ components of sea-ice velocity and the sea-ice speed. Because of the importance of these three variables in examining the movement of sea ice, for example in response to strong cyclones, they are requested as daily and monthly means. The specific request of sea-ice speed allows researchers to analyze the high-resolution movement of sea ice, for example, in comparison to the sub-daily speed reported by sea-ice drift buoys, without the error associated with vector averaging.

At priority 2, SIMIP requests more detailed variables describing sea-ice mass transports and the integrated forces on the ice cover. Hence, at this priority the data request includes the $x-$ and $y-$ components of the atmospheric and oceanic stress and the mass transport of sea ice. This then allows one to examine at first order to what degree the sea ice moves in response to atmospheric winds or in response to oceanic currents. Also the integrated measures sea-ice divergence and shear are requested at this level.

To allow for an even deeper understanding of the driving forces of sea-ice movement in large-scale model simulations, at priority 3, we request the individual terms of the force balance that determine the movement of the ice. These terms include for example the sea-surface tilt term, the Coriolis force, and the internal stresses. Because we only request those variables as monthly means, a true closure of the momentum balance will not be possible. However, the monthly mean values of these terms already allow one to identify the key processes that give rise to differences in CMIP6 sea-ice model simulations, which then in turn will hopefully allow improvements of the sea-ice dynamics in these models.





## 3.5 Integrated quantities

The final group of variables requested within SIMIP are primarily a service to the research community, consisting of integrated
quantities so often used in studies examining the evolution of sea ice that we felt it useful to make these quantities readily
available: total sea-ice area, sea-ice volume and sea-ice extent for the Northern and Southern hemispheres. In addition, we
request the areal and volume fluxes of sea ice through the four main outlets of the Arctic Ocean, namely Fram Strait, Bering
Strait, the Barents opening and the Canadian Archipelago. Note that for studies of sea-ice coverage, care must be taken in
using the non-linear diagnostic sea-ice extent (see Notz, 2014). In particular for pure model-intercomparison studies, where the
greater observational uncertainty of sea-ice area is irrelevant, sea-ice area should be the preferred diagnostic for the analysis
of sea-ice coverage. We hope that by providing these integrated variables directly as part of the model output data, a greater
number of researchers will be able to analyse the large-scale sea-ice evolution in both hemispheres, thus allowing for its ever
more complete understanding.

## 3.6 Observations

As outlined, SIMIP uses model simulations to understand the sea-ice evolution of the real world. This is done by linking the
model simulations to the real world through observations. Hence, SIMIP would not be possible without a reliable, wide range
of observational records that allow one to understand if CMIP6 model simulations capture the most important aspects of sea-ice
evolution as it also occurs in reality. For this reason, SIMIP does not only see itself as pure model-intercomparison exercise, but
also as a forum for identifying the best possible use of observations for the evaluation and improvement of model simulations.
For this purpose, SIMIP works closely with the National Snow and Ice Data Center (NSIDC), with other data centers and with
the observational community to maintain a detailed, up-to-date list of sea-ice related observational records that can be used by
any researcher to analyse the performance of sea-ice model simulations. We did not consider it useful to provide a snapshot of
this list here, given that it is quickly evolving and expanding. For details, please see http://www.climate-cryosphere.org/simip

## 4   Summary

SIMIP is a CMIP6-endorsed diagnostic MIP (Eyring et al., 2015). Its overarching aim is to improve our understanding of the
role of sea ice in the climate system. To achieve this goal, SIMIP requests no additional simulations, but instead asks for sea-ice
related model output following a newly developed data request. This model output allows researchers to analyse the three main
budgets that cover the evolution of sea ice, namely the heat budget, the momentum budget and the mass budget. This then
permits, for example, an analysis of the role of internal variability, external forcing, model tuning, and the formulation of the
sea-ice model for the quality of sea-ice simulations. The sea-ice variables that SIMIP requests are grouped into five categories,
namely (1) sea-ice state variables, (2) tendencies of sea-ice mass, (3) heat and freshwater fluxes, (4) sea-ice dynamics and (5)
integrated quantities. For each requested variable, we specify a priority that describes how crucial knowledge of this variable is



for our understanding of sea ice. Updates on SIMIP, including changes to the variable request and related observational datasets and publications will be listed at http://www.climate-cryosphere.org/simip.

**Data Availability**

5    All model output requested by SIMIP will be distributed through the Earth System Grid Federation (ESGF) with digital object identifiers (DOIs) assigned. As in CMIP5, the model output will be freely accessible through data portals after registration. In order to document CMIP6's scientific impact and enable ongoing support of CMIP, users are obligated to acknowledge CMIP6, the participating modelling groups, and the ESGF centres (see details on the CMIP Panel website at http://www.wcrp-climate.org/index.php/wgcm-cmip/about-cmip). Further information about the infrastructure supporting CMIP6, the different

10    CMIP6 MIPs, metadata describing the model output, model documentation, and the terms governing its use are provided by the WGCM Infrastructure Panel (WIP) in their contribution to this Special Issue.




## Appendix

In this appendix, we outline the SIMIP data request version 1.0 and the related request for model documentation. To account for possible long-term adjustments of this request, also for studies beyond CMIP6, an online version of this request is available at http://www.climate-cryosphere.org/simip. This website contains in particular a link to a spreadsheet version of the variable

list.

### Appendix A:  Model documentation request

In addition to the variable request, SIMIP requests that modeling groups provide documentation of their sea-ice model that allows a better interpretation of sea-ice simulations from individual CMIP6 models. For example, an assessment of the model simulations based on the use of thickness distributions or diagnostic salinity can provide insights into whether models that

include such advanced properties generally perform better than models that do not. Basic sea-ice model documentation was requested in past CMIPs, and SIMIP is contributing additional requests to fill gaps and include requests related to the SIMIP data request for CMIP6 (personal communication, Bryan Lawrence, March 2016).

We encourage all modeling groups to provide the requested model documentation information with as much detail as possible, and all scientists working with CMIP6 sea-ice output to make use of the model documentation to enhance the analysis

of the sea-ice simulations. A separate future publication in GMD will outline the sea-ice model documentation request for CMIP6.

As groups prepare the model output following the SIMIP request, the questions below should be kept in mind so they can easily be contributed to the upcoming more detailed request for sea-ice model documentation.

– Which combination of terms closes the mass and energy budgets in your model?

– Is your model missing any processes that relate to the requested sea-ice variables?

– Were any assumptions made in the calculation of diagnostic sea-ice variables we requested? If so, which ones?

– Does sea ice salinity impact the thermal properties of sea ice? (yes or no)

– Does your model use two different salinities for thermodynamic calculations and for the salt budget? (yes or no)

– Is the salinity used for ice-ocean exchanges variable or a constant? If constant, what is this constant?

– What kind of ice thickness distribution is used (if any)? How many categories and what are the category limits?

– How is the heat content of precipitation handled in your sea-ice model?

As discussed in section 2.1, SIMIP is also interested in assessing the tuning used in the CMIP6 models. In the model documentation request, the following questions will be asked related to the tuning that was done prior to the model code freeze for CMIP6:





– Which tuning knobs were used in tuning the sea-ice model?

– In what kind of simulations was the tuning done (control, transient 20th and 21st century, all of these)?

– What were the sea ice targets in the tuning effort?

**Appendix B: Some general remarks, including averaging etc.**

In the following appendices, we list all variables of the SIMIP data request version 1.0, grouped into the following five categories: (1) Sea-ice state variables, (2) tendencies of sea-ice mass, (3) heat and freshwater fluxes, (4) sea-ice dynamics and (5) integrated quantities.

Regarding the spatial storage of variables, for simplicity we request all variables to be stored on the model grid on which they actually are used during a model simulation. For most variables that SIMIP asks for, this will usually be the ocean grid.

However, for example individual atmospheric fluxes over the sea-ice covered part of a grid cell are often only evaluated in the atmosphere model, and should then simply be stored on the atmospheric grid.

Fractional coverages are evaluated either relative to the entire grid cell (e.g. for sea-ice concentration) or relative to the sea-ice covered part of the grid cell (e.g., meltpond area fraction). This is specified in detail in the following request. As a general rule, we have tried to follow the most wide-spread standard by which individual fractional coverages are usually represented

in sea-ice models.

Temporal averages are requested for almost all variables, except for a few variables related to sea-ice dynamics that must be stored as instantaneous values at some point during the averaging period. The standard averaging period for all variables is one month. For the most important variables, additional daily averages are requested, as identified in the following detailed data request.

For all variables that are proportional to area fraction, i.e. extensive variables such as volume, mass, or area fraction, a zero should always be averaged in for all time steps where no sea ice is present. This is because the extensive variables naturally approach zero as area fraction approaches zero.

For all variables that are not proportional to area fraction, i.e. intensive variables such as albedo, temperature, or heat flux, SIMIP requests the area-weighted average. Hence, all time samples with non-zero sea-ice fraction are first multiplied by area

fraction, then summed, and then divided by the sum of the area fractions. Ice-free grid-cells at any point throughout the averaging period should be treated as missing values and the averaging should only be carried out for those periods where sea ice is present. For continuously ice-free grid cells, missing values should be reported. This is because the intensive variables do not necessarily approach zero as area fraction approaches zero.

To report grid-cell averages for multi-category models, the properties of the individual categories should be averaged to a

single value for each time step by calculating the area-weighted average across all categories. The single value thus obtained for each time step should then be used for all further processing of model output. The only exception to this rule are variables that specifically ask for values for individual categories.





## Appendix C: State variables

The most fundamental set of variables that SIMIP requests are those variables that describe the actual state of the sea ice cover. This set of variables allows one to examine for example how much sea ice there actually is in a certain region, how thick the sea-ice cover is, whether there is snow on sea ice and how densely the sea-ice cover is packed. Knowledge of the temporal

evolution of these parameters then allows one to examine, for example, the seasonal cycle of the sea-ice cover or its long-term evolution. In addition, SIMIP also requests some fundamental thermodynamic quantities such as the surface temperature of the ice or the heat content of the ice cover, which allows one to leading order close energy budgets related to sea ice. Because of the fundamental nature of these quantities, many of them are requested at priority 1. At priority 2, SIMIP requests variables that are slightly less central, but still relevant for many researchers, for example surface albedo or heat content of the ice cover.

At priority 3, finally, SIMIP requests variables that will only be available from advanced model formulations that include for example a melt-pond scheme or a scheme to interactively calculate the bulk salinity of the ice cover.

### C1  Priority 1

#### C1.1  Fraction of time steps with sea ice (*sitimefrac*)

Fraction of time steps of the averaging period during which sea ice is present (siconc >0 ) in a grid cell. This is in particular

useful for the SIMIP standard averaging period of a month, since many researchers will only analyse those months where sea ice was present in a particular grid cell for the entire averaging period. Requested as daily and monthly average.

#### C1.2  Sea-ice area fraction (*siconc*)

Areal fraction of a given grid cell that is covered by sea ice, independent of the thickness of that ice. By definition, this variable can only have values between 0 (no sea ice at all) and 1 (fully covered by sea ice). Requested as daily and monthly average.

#### C1.3  Sea-ice mass per area (*simass*)

Total mass of sea ice divided by the entire area of a grid cell. Mass is the truly conserved quantity, so we prioritise requesting sea-ice mass over requesting sea-ice volume.

#### C1.4  Sea-ice thickness (*sithick*)

Thickness of sea ice averaged over the ice-covered part of a given grid cell. This variable hence describes the actual thickness

of the sea ice, which in the context of the heat budget is for example necessary to analyse the heat flux through the ice. From a sea-ice perspective, this real (or floe) thickness is a more meaningful variable to store than the so-called equivalent sea-ice thickness that was used in previous CMIPs, which is defined as the sea-ice volume divided by the area of the entire grid cell. While use of equivalent thickness (i.e., volume) is useful from the perspective of an ocean model, it is not meaningful in a sea-ice context. Indeed, in our experience users often assumed that such equivalent thickness was the actual thickness of the





sea ice, which is why we request the actual sea-ice thickness as a new variable for CMIP6. It should be directly accessible in any sea-ice model, but can otherwise be calculated by dividing the total sea-ice volume by the sea-ice area. Requested as daily and monthly average.

### C1.5 Snow area fraction (*sisnconc*)

Area fraction of the sea-ice surface that is covered by snow. In many models that do not explicitly resolve an areal fraction of snow, this variable will always be either 0 or 1.

### C1.6 Snow mass per area (*sisnmass*)

Total mass of snow on sea ice divided by the entire area of a grid cell. This then allows one to analyse the storage of latent heat in the snow, and to calculate the snow-water equivalent.

### C1.7 Snow thickness (*sisnthick*)

Thickness of snow averaged over the snow-covered part of the sea ice. It hence describes the actual thickness of the snow, which in the context of the heat budget is for example necessary to analyse the heat flux through the ice and snow. This thickness is usually directly available within the model formulation. It can also be derived by dividing the total volume of snow through the area of the snow. Requested as daily and monthly average.

### C1.8 Surface temperature (*sitemptop*)

Mean surface temperature of the sea-ice covered part of the grid cell. Wherever snow covers the ice, the surface temperature of the snow is used for the averaging, otherwise the surface temperature of the ice is used. Requested as daily and monthly average.

### C1.9 Sea-ice volume per area (*sivol*)

This is also known as the equivalent thickness of sea ice, which is calculated by dividing the volume of sea ice by the entire grid area. This measure used to simply be called ice thickness in previous CMIPs, which gave rise to some confusion for users expecting this variable to describe actual thickness. Since ice mass is more general than volume, this variable is somehow obsolete and primarily part of SIMIP for these historical reasons.

### C2 Priority 2

### C2.1 Temperature at snow-ice interface (*sitempsnic*)

Report surface temperature of ice where snow thickness is zero





### C2.2 Temperature at ice-ocean interface (*sitempbot*)

Report temperature at interface, NOT temperature within lowermost sea-ice model layer

### C2.3 Age of sea ice (*siage*)

Age of sea ice since its formation in open water

### C2.4 Sea-ice or snow albedo (*sialb*)

Mean surface albedo of entire ice-covered part of grid cell

### C2.5 Sea-ice freeboard (*sifb*)

Mean height of sea-ice surface (=snow-ice interface when snow covered) above sea level

### C2.6 Sea-ice heat content per unit area (*sihc*)

Heat content of all ice in grid cell divided by total grid-cell area. This includes both the latent and sensible heat content contribution. Water at 0 °C is assumed to have a heat content of 0 J. This variable does not include heat content of snow, but does include heat content of brine. Heat content is always negative, since both the sensible and the latent heat content of ice are less than that of water

### C2.7 Snow-heat content per unit area (*sisnhc*)

Heat-content of all snow in grid cell divided by total grid-cell area. This includes both the latent and sensible heat content contribution. Snow-water equivalent at 0 °C is assumed to have a heat content of 0 J. Does not include heat content of sea ice.

## C3 Priority 3

### C3.1 Sea-ice area fractions in thickness categories (*siitdconc*)

Area fraction of grid cell covered by each ice-thickness category (vector with one entry for each thickness category starting from the thinnest category, netcdf file should use thickness bounds of the categories as third coordinate axis)

### C3.2 Sea-ice thickness in thickness categories (*siitdthick*)

Actual (floe) thickness of sea ice in each category (NOT volume divided by grid area), (vector with one entry for each thickness category starting from the thinnest category, netcdf file should use thickness bounds of categories as third coordinate axis)



### C3.3 Snow area fractions in thickness categories (*siitdsnconc*)

Area fraction of grid cell covered by snow in each ice-thickness category (vector with one entry for each thickness category starting from the thinnest category, netcdf file should use thickness bounds of the categories as third coordinate axis)

### C3.4 Snow thickness in thickness categories (*siitdsnthick*)

5 Actual thickness of snow in each category (NOT volume divided by grid area), (vector with one entry for each thickness category starting from the thinnest category, netcdf file should use thickness bounds of categories as third coordinate axis)

### C3.5 Mass of salt in sea ice per area (*sisaltmass*)

Total mass of all salt in sea ice divided by grid-cell area. Sometimes, models implicitly or explicitly assume a different salinity of the ice for thermodynamic considerations than they do for closing the salt budget with the ocean. In these cases, the total 10 mass of all salt in sea ice should be calculated from the salinity value used in the calculation of the salt budget.

### C3.6 Meltpond area fraction (*simpconc*)

Area fraction of sea-ice surface that is covered by melt ponds

### C3.7 Meltpond depth (*simpthick*)

Volume of water in meltponds divided by meltpond covered area

### 15 C3.8 Thickness of refozen ice on melt pond (*simprefrozen*)

Volume of refrozen ice on melt ponds divided by meltpond covered area

### C3.9 Ridged ice area fraction (*sirdgconc*)

Area fraction of sea-ice surface that is covered by ridged sea ice

### C3.10 Ridged ice thickness (*sirdgthick*)

20 Total volume of ridged sea ice divided by area of ridges

### C3.11 Sea ice salinity (*sisali*)

Mean sea-ice salinity of all sea ice in grid cell. Sometimes, models implicitly or explicitly assume a different salinity of the ice for thermodynamic considerations than they do for closing the salt budget with the ocean. In these cases, the mean salinity used in the calculation of the salt budget should be reported





## Appendix D: Tendencies of sea-ice mass and area fraction

While the sea-ice state variables already allow one to calculate how much the amount of sea ice in a certain grid cell is changing from one averaging period to the next, such estimate does not allow one to infer why precisely the sea-cover is changing the way it does, i.e. whether a given mass change is driven by dynamics or by thermodynamics. Therefore, SIMIP requests a rather

detailed list of variables that describe where the amount of sea ice is changing. These variables are all requested at priority 2, since they are usually used for the in-depth analysis of the simulated sea-ice cover. These variables should all be readily available from any modern sea-ice model. All tendencies are negative for decreasing mass.

### D2    Priority 2

#### D2.1    sea-ice area fraction change from thermodynamics (*sidconcth*)

Total change in sea-ice area fraction through thermodynamic processes

#### D2.2    sea-ice area fraction change from dynamics (*sidconcdyn*)

Total change in sea-ice area fraction through dynamics-related processes (advection, divergence...)

#### D2.3    sea-ice mass change from thermodynamics (*sidmassth*)

Total change in sea-ice mass from thermodynamic processes divided by grid-cell area

#### D2.4    sea-ice mass change from dynamics (*sidmassdyn*)

Total change in sea-ice mass through dynamics-related processes (advection,... divided by grid-cell area)

#### D2.5    sea-ice mass change through growth in supercooled open water (aka frazil) (*sidmassgrowthwat*)

The rate of change of sea-ice mass due to sea ice formation in supercooled water (often through frazil formation) divided by grid-cell area. Together, sidmassgrowthwat and sidmassgrowthbot should give total ice growth from sea water. Always positive

or zero.

#### D2.6    sea-ice mass change through basal growth (*sidmassgrowthbot*)

The rate of change of sea-ice mass due to vertical growth of existing sea ice at its base divided by grid-cell area. Note that this number is always positive or zero, since sea-ice melt is collected in *sidmassmeltbot*. This is to account for differential growth and melt in models with a sub-grid scale ice-thickness distribution.

#### D2.7    sea-ice mass change through snow-to-ice conversion (*sidmasssi*)

The rate of change of sea-ice mass due to transformation of snow to sea ice divided by grid-cell area. Always positive or zero.





### D2.8 sea-ice mass change through evaporation and sublimation (*sidmassevapsubl*)

The rate of change of sea-ice mass change through evaporation and sublimation divided by grid-cell area

### D2.9 sea-ice mass change through surface melting (*sidmassmelttop*)

The rate of change of sea-ice mass through melting at the ice surface divided by grid-cell area. This number is independent of
the actual fate of the melt water, and will hence include all sea-ice melt water that drains into the ocean and all sea-ice melt
water that is collected by a melt-pond parameterisation. Always negative or zero.

### D2.10 sea-ice mass change through bottom melting (*sidmassmeltbot*)

The rate of change of sea-ice mass through melting/dissolution at the ice bottom divided by grid-cell area. Note that this number
is always zero or negative, since sea-ice growth is collected in *sidmassgrowthbot*. This is to account for differential growth and
melt in models with a sub-grid scale ice-thickness distribution.

### D2.11 sea-ice mass change through lateral melting (*sidmasslat*)

The rate of change of sea-ice mass through lateral melting/dissolution divided by grid-cell area (report 0 if not explicitly
calculated thermodynamically). Always negative or zero.

### D2.12 snow mass change through snow fall (*sndmasssnf*)

Mass of solid precipitation falling onto sea ice divided by grid-cell area. Always positive or zero.

### D2.13 snow mass change through melt (*sndmassmelt*)

The rate of change of snow mass through melt divided by grid-cell area. Always negative or zero.

### D2.14 snow mass change through sublimation (*sndmasssubl*)

The rate of change of snow mass through sublimation divided by grid-cell area

### D2.15 snow mass change through advection by sea-ice dynamics (*sndmassdyn*)

The rate of change of snow mass through advection with sea ice divided by grid-cell area

### D2.16 snow mass change through snow-to-ice conversion (*sndmasssi*)

The rate of change of snow mass due to transformation of snow to sea ice divided by grid-cell area. Always negative or zero.





### D2.17 snow mass change through wind drift of snow (*sndmasswindrif*)

The rate of change of snow mass due to wind-driven transport into the ocean

### Appendix E: Heat and freshwater fluxes (all only for sea-ice fraction of grid cell, downward always positive)

To understand the drivers of the sea-ice tendencies introduced in the previous section, SIMIP requests storage of the actual heat
fluxes that gave rise to the tendencies in sea-ice mass. In contrast to previous CMIPs, all fluxes are to be evaluated over the
ice-covered part of the grid cell. In earlier CMIPs, the fluxes were usually just provided as a grid-cell average, including the
fluxes over open water. This made it impossible to analyse the heat budget of sea ice, which is why SIMIP now requests these
fluxes over sea ice. In many coupled models, the individual fluxes over sea ice are only available on the atmospheric grid since
only integrated net fluxes are passed on to the ocean model. In this case, the fluxes over sea ice should simply be stored on the
atmospheric grid.

In addition to the analysis of the heat fluxes, also the freshwater fluxes are an important driver for the interaction of sea ice
with the climate system of the Earth. To understand the magnitude of these freshwater fluxes, SIMIP requests storage of both
the salt flux and of the freshwater flux from sea ice melt or growth.

The sign convention is generally positive downward. However, to remain consistent with the NetCDF Climate and Forecast
(CF) convention, upward fluxes that carry the term "upward" in their name are positive upward, as detailed in the following.

### E2 Priority 2

#### E2.1 Downwelling shortwave flux over sea ice (*siflswdtop*)

The downwelling shortwave flux from the atmosphere to the sea-ice surface. Always positive or zero.

#### E2.2 Upward shortwave flux over sea ice (*siflswutop*)

The upward shortwave flux from the sea-ice surface to the atmosphere. Always positive or zero.

#### E2.3 Downwelling shortwave flux under sea ice (*siflswdbot*)

The downwelling shortwave flux underneath sea ice, i.e. the amount of shortwave radiation that penetrates the sea ice. Always
positive or zero.

#### E2.4 Downwelling longwave flux over sea ice (*sifllwdtop*)

The downwelling longwave flux from the atmosphere to the sea-ice surface. Always positive or zero.





### E2.5    Upward longwave flux over sea ice (*sifllwutop*)

The upward longwave flux from the sea-ice surface to the atmosphere. Always positive or zero.

### E2.6    Net sensible heat flux over sea ice (*siflsenstop*)

The net sensible heat flux over sea ice. Positive for a downward heatflux.

### E2.7    Net latent heat flux over sea ice (*sifllatstop*)

The net latent heat flux over sea ice. Positive for a downward heatflux.

### E2.8    Net sensible heat flux under sea ice (*siflsensupbot*)

The net sensible heat flux under sea ice from or to the ocean. Per sign convention, heat from the ocean is counted as negative since it describes an upward heat flux.

### E2.9    Net conductive heat flux in ice at the surface (*siflcondtop*)

The net heat conduction flux at the ice surface, i.e. the conductive heat flux from the center of the uppermost vertical sea-ice grid box to the surface of the sea ice. Positive for a downward heatflux.

### E2.10    Net conductive heat fluxes in ice at the bottom (*siflcondbot*)

The net heat conduction flux at the ice base, i.e. the conductive heat flux from the center of the lowermost vertical sea-ice grid box to the bottom of the sea ice. Positive for a downward heatflux.

### E2.11    Rainfall rate over sea ice (*sipr*)

Mass of liquid precipitation falling onto sea ice divided by grid-cell area. If the rain is directly put into the ocean, it should not be counted towards sipr. Always positive or zero.

### E2.12    Salt flux from sea ice (*siflsaltbot*)

Total flux of salt from water into sea ice divided by grid-cell area; salt flux is upward (negative) during ice growth when salt is embedded into the ice and downward (positive) during melt when salt from sea ice is again released to the ocean

### E2.13    Freshwater flux from sea ice (*siflfwbot*)

Total flux of fresh water from water into sea ice divided by grid-cell area; This flux is negative during ice growth (liquid water mass decreases, hence upward flux of freshwater), positive during ice melt (liquid water mass increases, hence downward flux of freshwater)





## Appendix F: Sea-ice dynamics

The variables that SIMIP requests for sea-ice dynamics are needed to understand how and why sea ice moves horizontally. The most important parameters there are obviously the actual velocities, which are requested on the native model grid. In addition, the sea-ice speed is requested, which allows one to account fo possible back-and-forth movement of the ice during the averaging period. Because of their importance, these variables are requested at priority 1 and at daily resolution.

At lower priority, SIMIP requests primarily those variables that allow one to examine the various forces that are responsible for the actual sea-ice movement. These are at priority 2 the total atmospheric and the total ocenic stress and the strength of the sea-ice cover. At priority 3, further details on the actual forces are requested.

Note that four variables are requested as instantenous values. These are the divergence and the maximum shear of the sea-ice velocity field, and the average normal stress and the maximum shear stress in the sea ice. These variables can only usefully be analysed if stored simultaneously as instantenous values at some point during the averaging period.

### F1 Priority 1

#### F1.1 X-component of sea-ice velocity (*siu*)

X-component of sea-ice velocity. Requested as daily and monthly average.

#### F1.2 Y-component of sea-ice velocity (*siv*)

Y-component of sea-ice velocity. Requested as daily and monthly average.

#### F1.3 Sea-ice speed (*sispeed*)

Speed of ice (i.e. mean absolute velocity) to account for back-and-forth movement of the ice. Requested as daily and monthly average.

### F2 Priority 2

#### F2.1 X-component of sea-ice mass transport (*sidmasstranx*)

Includes transport of both sea ice and snow by advection

#### F2.2 Y-component of sea-ice mass transport (*sidmasstrany*)

Includes transport of both sea ice and snow by advection

#### F2.3 X-component of atmospheric stress on sea ice (*sistrxdtop*)

X-component of atmospheric stress on sea ice





### F2.4 Y-component of atmospheric stress on sea ice (*sistrydtop*)

Y-component of atmospheric stress on sea ice

### F2.5 X-component of ocean stress on sea ice (*sistrxubot*)

X-component of ocean stress on sea ice

### F2.6 Y-component of ocean stress on sea ice (*sistryubot*)

Y-component of ocean stress on sea ice

### F2.7 Compressive sea ice strength (*sicompstren*)

Computed strength of the ice pack, defined as the energy (J m-2) dissipated per unit area removed from the ice pack under compression, and assumed proportional to the change in potential energy caused by ridging. For Hibler-type models, this is $P(= P^* \cdot h \cdot \exp(-C(1-A)))$

### F2.8 Divergence of the sea-ice velocity field (*sidivvel*)

Divergence of sea-ice velocity field (first shear strain invariant). Requested as instantaneous value.

### F2.9 Maximum shear of sea-ice velocity field (*sishevel*)

Maximum shear of sea-ice velocity field (second shear strain invariant). Requested as instantaneous value.

## F3 Priority 3

### F3.1 Atmospheric drag coefficient (*sidragtop*)

Atmospheric drag coefficient that is used to calculate the atmospheric momentum drag on sea ice

### F3.2 Ocean drag coefficient (*sidragbot*)

Oceanic drag coefficient that is used to calculate the oceanic momentum drag on sea ice

### F3.3 Sea-surface tilt term in force balance (x-component) (*siforcetiltx*)

X-component of force on sea ice caused by sea-surface tilt

### F3.4 Sea-surface tilt term in force balance (y-component) (*siforcetilty*)

Y-component of force on sea ice caused by sea-surface tilt





### F3.5   Coriolis force term in force balance (x-component) (*siforcecoriolx*)

X-component of force on sea ice caused by coriolis force

### F3.6   Coriolis force term in force balance (y-component) (*siforcecorioly*)

Y-component of force on sea ice caused by coriolis force

### F3.7   Internal stress term in force balance (x-component) (*siforceintstrx*)

X-component of force on sea ice caused by internal stress (divergence of sigma)

### F3.8   Internal stress term in force balance (y-component) (*siforceintstry*)

Y-component of force on sea ice caused by internal stress (divergence of sigma)

### F3.9   Average normal stress in sea ice (*sistresave*)

Average normal stress in sea ice (first stress invariant). Requested as instantaneous value.

### F3.10   Maximum shear stress in sea ice (*sistremax*)

Maximum shear stress in sea ice (second stress invariant). Requested as instantaneous value.

### Appendix G:  Integrated measures

Much of the analysis of sea ice in climate research is concerned with integrated measures such as total hemispheric sea-ice area or sea-ice volume. Within SIMIP, we hence consider it useful to have these measures directly available, and request them at priority 2. Note that care should be taken in the use of sea-ice extent, because its non-linear behaviour can cause substantial artefacts both regarding the spatial and the temporal evolution of the sea-ice cover (see Notz, 2014). In particular in pure model intercomparison studies, where the higher observational uncertainty of sea-ice area is irrelevant, the linear metric sea-ice area should be preferred.

## G2   Priority 2

### G2.1   Sea-ice area North (*siarean*)

total area of sea ice in the Northern hemisphere

### G2.2   Sea-ice area South (*siareas*)

total area of sea ice in the Southern hemisphere



### G2.3 Sea-ice volume North (*sivoln*)

total volume of sea ice in the Northern hemisphere

### G2.4 Sea-ice volume South (*sivols*)

total volume of sea ice in the Southern hemisphere

### G2.5 Sea ice extent North (*siextentn*)

Total area of all Northern-Hemisphere grid cells that are covered by at least 0.15 areal fraction of sea ice

### G2.6 Sea ice extent South (*siextents*)

Total area of all Southern-Hemisphere grid cells that are covered by at least 0.15 areal fraction of sea ice

### G2.7 Sea-ice-mass flux through straits (*simassacrossline*)

Net (sum of transport in all directions) sea-ice-mass transport through the following four passages, positive into the Arctic Ocean. Note that the definitions of the passages are for SIMIP purposes just meant as default values as given by the physical ocean MIP described in Griffies et al. (2016). Individual models might chose slightly different definitions as given by their grid geometry.

1. Fram Strait = (11.5°W,81.3°N) to (10.5°E,79.6°)
2. Canadian Archipelego = (128.2°W,70.6°N) to (59.3°W,82.1°)
3. Barents opening = (16.8°E,76.5°N) to (19.2°E,70.2°N)
4. Bering Strait = (171°W,66.2°N) to (166°W,65°N)

### G2.8 Sea-ice area flux through straits (*siareaacrossline*)

Net (sum of transport in all directions) sea-ice-area transport through the following four passages, positive into the Arctic Ocean. Note that the definitions of the passages are for SIMIP purposes just meant as default values as given by the physical ocean MIP described in Griffies et al. (2016). Individual models might chose slightly different definitions as given by their grid geometry.

1. Fram Strait = (11.5°W,81.3°N) to (10.5°E,79.6°)
2. Canadian Archipelego = (128.2°W,70.6°N) to (59.3°W,82.1°)
3. Barents opening = (16.8°E,76.5°N) to (19.2°E,70.2°N)
4. Bering Strait = (171°W,66.2°N) to (166°W,65°N)





### G2.9 Snow mass flux through straits (*snmassacrossline*)

Net (sum of transport in all directions) snow mass transport through the following four passages, positive into the Arctic Ocean. Note that the definitions of the passages are for SIMIP purposes just meant as default values as given by the physical ocean MIP described in Griffies et al. (2016). Individual models might chose slightly different definitions as given by their grid geometry.

5  1. Fram Strait = (11.5°W,81.3°N) to (10.5°E,79.6°)

2. Canadian Archipelego = (128.2°W,70.6°N) to (59.3°W,82.1°)

3. Barents opening = (16.8°E,76.5°N) to (19.2°E,70.2°N)

4. Bering Strait = (171°W,66.2°N) to (166°W,65°N)

*Acknowledgements.* We thank the international sea-ice community for their input and guidance during the design of SIMIP, most notably
10  during the Sea Ice and Climate Modeling workshop in Reading on 26th September 2014. We are very grateful to Martin Juckes for his tremendous help and support throughout our work on the design of SIMIP, and to David Bailey, Helmuth Haak and Alex West for their helpful comments and feedback. We thank WCRP CliC for their logistical and financial support, which are essential for SIMIP.



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
