# Peer review of "Sea Ice Model Intercomparison Project (SIMIP): Understanding sea ice through climate-model simulations"

_Geoscientific Model Development, 2016_

## Short Comment (SC1) · 13 Apr 2016

Dear authors,

In agreement with the CMIP6 panel members, the Executive editors of GMD would like to establish a common naming convention for the titles of the CMIP6 experiment description papers.

The title of CMIP6 papers should include both the acronym of the MIP, and CMIP6, so that it is clear this is a CMIP6-Endorsed MIP.

Additionally, we strongly recommend to add a version number to the MIP descrip-tion. The reason for the version numbers is so that the MIP protocol can be updated later, normally in a second short paper outlining the changes. See, for example: http://www.geosci-model-dev.net/special_issue11.html,

Good formats for the title include:

'XYZMIP (v1.0) contribution to CMIP6: Name of project'

or

'Name of Project (XYZMIP v1.0) contribution to CMIP6'

If you want to include a more descriptive title, the format could be along the lines of,

'XYZMIP (v1.0) contribution to CMIP6: Name of project - descriptive title'

or

'Name of Project (XYZMIP v1.0) contribution to CMIP6: descriptive title.'

When you revise your manuscript, please correct the title of your manuscript accordingly.

Yours,

Astrid Kerkweg

---

## Referee Comment (RC1) · D. Bailey (Referee) · 16 May 2016

This manuscript describes the overall motivation and description of the requested documentation and variables for the sea ice part of the Coupled Model Intercomparison Project (CMIP). Overall, I feel this is a very good descriptive paper and should be considered for publication subject to a few minor revisions which I have listed here.

1. The concept of "extensive" and "intensive" variables is still a bit confusing to me. I can think of examples such at top melting on sea ice (sidmassmelttop) that I could argue either way. I would request that in the appendices C-F the authors add the label of extensive or intensive to the variables, so that we in the community can be on the same page.

2. The concept of sea ice freeboard is interesting and challenging even for the remote sensing community. I believe more description should be added here so as to make it clear what satellite product(s) we would be comparing the models against.

3. Another issue is that most models that I am aware of do not keep track of terms for the sea ice and snow on top of sea ice separately. I would recommend that the following be expanded/clarified as to what is preferred: sidmassevapsubl and sndmassdyn. I know that in the CICE model we do not separate these for sea ice/snow.

Note that while I checked Excellent for scientific reproducibility, I believe this should actually be N/A.

---

## Referee Comment (RC2) · Anonymous Referee #2 · 18 May 2016

This paper summarizes the goals and design of the CMIP6-endorsed Sea Ice Model Intercomparison (SIMIP) project, which is a diagnostic MIP that requests specific sea-ice related outputs to be saved so that a detailed assessment of sea ice variability, its response to external forcing and its influence on climate can be made.

The questions that can be addressed with the suggested outputs are fundamental, not only for sea ice scientists but for a wide range of climate scientists and stakeholders. Indeed previous analysis of climate simulations highlighted large differences between model simulations of past sea ice evolution and observations. They also indicated a large model spread and hence a large uncertainty in future projections of sea ice. The diagnostics that will be possible with the outputs requested by SIMIP will allow

to better identify whether model differences arise from internal variability, from different tuning approaches, or/and from missing processes. It will also allow an improved understanding of the processes that govern sea ice evolution and its interaction with the atmosphere above and the ocean below. Very few sea-ice variables were saved in previous CMIP experiments and a rigorous analysis of the heat budget of sea ice was not possible. The data request detailed in this document includes new variables so that such budgets can be computed in upcoming CMIP6 simulations. An interesting suggestion is also to request that modeling centers provide a detailed documentation of their sea ice model and of the tuning approach they follow. This information is currently missing for many sea ice models and I think it is very important to better understand the differences between models and to help young scientists who start in this field.

The expected outcome of this project is hence a better understanding of the drivers of sea ice internal variability, more reliable projections of sea ice changes and sea ice related climate changes, a better estimate of models uncertainty and eventually possible improvement of systematic biases in climate models.

The authors made an outstanding effort to come up with such a detailed description of the requested variables. The document gives a detailed description of the requested data by dividing them into 5 groups, each describing a specific aspect of sea ice evolution and associated with a priority 1 to 3 .The document is very clear even for a non-expert in sea ice modeling. Detailed guidance is given on every requested variable, including its definition and what type of analysis it can be used for. Thus, I strongly recommend the publication of this paper and I only have few minor changes to suggest as detailed below.

Detailed comments:

-Appendix E. p.20. l.3: I find it confusing to put "downward always positive" in the title here and have fluxes defined positive upward l.15, l.20 and in the following page. I am fine with this convention but I suggest phrasing it differently in the title l.3 e.g. "usually

positive downward except when the term upward appears in the name description"

-Appendix F p.22 l.4: To me the request of sea ice speed is redundant with the x and y components of sea ice velocity. Why can't we estimate sea ice speed only from siu and siv ? In addition, I don't understand what is meant by " to account for back-and-forth movement of sea ice". Isn't the speed an absolute value that is positive by definition and hence does not provide any information about the direction? Please clarify the explanation and remove sispeed from the requested variables if it can be estimated from siu and siv.

-p.23, l.10: P, h, C and A are not defined. I guess that sea-ice experts who use a Hibler model would understand but it would be useful to add at least a reference where this term is clearly defined.

Typos: -p.10, l.12, I guess the authors meant to use the word "even" instead of "ever" -p.17, l.15: "refozen" should be replaced by "refrozen" -p.18, l.16 : the parenthesis should be closed after "advection..."

---

## Short Comment (SC2) · 8 Jun 2016

**Comments from CMIP Panel**

The CMIP Panel is undertaking a review of the CMIP6 GMD special issue papers to ensure a level of consistency among the invited contributions, also in answering the key questions that were outlined in our request to submit a paper to all co-chairs of CMIP6-Endorsed MIPs. We very much welcome the important contribution from SIMIP to the CMIP6 special issue, below are a few comments:

Please ensure that the title of your paper includes both the acronym of the MIP, and CMIP6, so that it is clear this is a CMIP6-Endorsed MIP.

[Figure]

Please consistently use the term *'CMIP6-Endorsed MIPs'* when you refer to other MIPs that are endorsed by CMIP6 (e.g. p2, l21; p10, l25)

Please ensure consistency of the experiment names and abbreviations with the CMIP6 overview paper (Eyring et al., 2016): for example p5, l10: replace 'control simulation' with 'pre-industrial control simulation'.

p5, l10: while the *piControl* and the CMIP6 historical simulations are specified as experiments from which the model output defined in SIMIP is requested, there is no mentioning of the Diagnostic, Evaluation and Characterization of Klima (DECK) experiments. Please specify whether the output should be collected also from the other CMIP DECK experiments (i.e. *amip, abrupt-4xCO2* and *1pctCO2*) and if so, why.

p5, l 10: is it necessary to collect all output from all CMIP6-Endorsed MIP experiments or can this be partly reduced to priority 1 output for example? Please could you be specific in the paper?

p2, l28: please replace *'CMIP6 data call'* with *'CMIP6 data request'* and you could refer to the invited contribution to this special issue or the website of the CMIP6 data request.

p10, l15ff: any plans to contribute or encourage the contribution of observations that could be used to evaluate the proposed experiments to obs4MIPs?

Appendix A: Model documentation request (p.12): detailed model documentation including information on tuning is clearly important. However, this information should be collected as part of the Earth System Documentation (ES-DOC) activity (see http://es-doc.org) rather than in a separate effort. Please ensure the information that SIMIP requires is communicated to the ES-DOC group.

Appendix B (p 13, l8ff). We agree it is best to collect all variables on the native model grids. However, some additional information from the models is required to allow re-gridding of the data to a common grid. OMIP is proposing a weights file that model

groups should provide to enable regridding from the native grid to one or two CMIP6 standard grids. Please refer to Griffies et al. (2016) and follow the same proceduce for sea ice requests.

Appendices C-G: This is a very helpful overview of the variables requested by SIMIP. It would be nice to identify for each variable whether this is a variable that can (at least in principle) be evaluated with observations. Are simulators such as the COSP simulator required for any model-observation comparisons?

References:

Eyring, V., Bony, S., Meehl, G. A., Senior, C. A., Stevens, B., Stouffer, R. J., and Taylor, K. E.: Overview of the Coupled Model Intercomparison Project Phase 6 (CMIP6) experimental design and organization, Geosci. Model Dev., 9, 1937-1958, doi:10.5194/gmd-9-1937-2016, 2016.

Griffies, S. M., Danabasoglu, G., Durack, P. J., Adcroft, A. J., Balaji, V., Böning, C. W., Chassignet, E. P., Curchitser, E., Deshayes, J., Drange, H., Fox-Kemper, B., Gleckler, P. J., Gregory, J. M., Haak, H., Hallberg, R. W., Hewitt, H. T., Holland, D. M., Ilyina, T., Jungclaus, J. H., Komuro, Y., Krasting, J. P., Large, W. G., Marsland, S. J., Masina, S., McDougall, T. J., Nurser, A. J. G., Orr, J. C., Pirani, A., Qiao, F., Stouffer, R. J., Taylor, K. E., Treguier, A. M., Tsujino, H., Uotila, P., Valdivieso, M., Winton, M., and Yeager, S. G.: Experimental and diagnostic protocol for the physical component of the CMIP6 Ocean Model Intercomparison Project (OMIP), Geosci. Model Dev. Discuss., doi:10.5194/gmd-2016-77, in review, 2016.

With many thanks for your ongoing efforts in the CMIP6 process.

The CMIP Panel

---

## Author Comment (AC1) · 7 Jul 2016

Dear executive editor,

thank you very much for pointing out that the title should include a direct reference to CMIP6. In a revised version of this paper, we will change the title accordingly.

Best regards.
* * *

---

## Author Comment (AC2) · 7 Jul 2016

Dear David,

thank you very much for your helpful and constructive feedback. In a revised version of this paper, we will indicate for each variable whether it should be treated as extensive or intensive, and we will clarify how freeboard and surface fluxes should be calculated.

Best regards.

---

## Author Comment (AC3) · 7 Jul 2016

We are very grateful for your positive and helpful evaluation of our manuscript. In a revised version, we will address all the issues you raise:

1. We will clarify the sign convention

2. We will explain why we request sea-ice speed. This is to account for the movement of sea ice on time scales below one day, which is the shortest period on which velocities are requested. For example, if tides cause the sea ice to move in one direction for 6 hours and then in the opposite direction for the following 6 hours, x and y components of daily velocities might be close to zero, while speed is actually non zero.

[Figure]

3. We will clarify the variables in the Hibler model

4. We will correct all issues with grammar and typos. Thanks for spotting them.

———————————————

---

## Author Comment (AC4) · 7 Jul 2016

Dear Veronika,

thank you very much for your careful reading and helpful evaluation of our manuscript. In a revised version, we will ensure that the issues that you raise are addressed as follows:

1. The title will include a reference to CMIP6.

2. We will consistently use the term 'CMIP6-Endorsed MIPs' when referring to other MIPs

3. We will ensure consistency of experiment names and abbreviations with the CMIP6 overview paper.

4. Regarding the experiments from which SIMIP output is requested, we will ensure that the paper becomes consistent with our SIMIP data request and explain the underlying reasoning.

5. Regarding the degree to which output is required from MIP experiments: We are hesitant in being to restrictive for the time being, since in our experience significant frustration can arise from missing output that was not deemed necessary when the experiments were run. To reflect this issue in our paper, we will differentiate between "recommended" output and "required" output.

6. We will replace 'CMIP6 data call' with 'CMIP6 data request' and add a reference.

7. Regarding observations, we will have two workshops over the next couple of months to discuss and encourage possible contributions of observations to obs4MIPs. As we assume that this will remain an ongoing effort, we will document these developments on the SIMIP homepage rather than in the paper itself.

8. We will ensure that the SIMIP documentation request becomes part of the ES-DOC activity.

9. We will request individual groups to provide a weights file to allow for the easy processing of gridded data and will add the OMIP paper as a reference.

10. Regarding model evaluation, we will document the observational requirements on our SIMIP homepage.

Best regards.
* * *

---

## Author Response (AR1)

**Reply to reviews**

We are very grateful for the very helpful comments that we have received in the online discussion. We thank both reviewers, the editor and the CMIP6 panel for the insight and time they spent on evaluating our manuscript.

In the revised version, we have addressed all comments as indicated in the following.

**Comments by Astrid Kerkweg**

*1. In agreement with the CMIP6 panel members, the Executive editors of GMD would like to establish a common naming convention for the titles of the CMIP6 experiment description papers. The title of CMIP6 papers should include both the acronym of the MIP, and CMIP6, so that it is clear this is a CMIP6-Endorsed MIP.*
**Changes made:** The title now reads: "The CMIP6 Sea Ice Model Intercomparison Project (SIMIP): Understanding sea ice through climate-model simulations"

*2. Additionally, we strongly recommend to add a version number to the MIP description. The reason for the version numbers is so that the MIP protocol can be updated later, normally in a second short paper outlining the changes.*
**Reply:** We have received additional guidance from the CMIP6 panel that specified that such version number is not required. We prefer to follow this advise and to not include a version number.

**Comments by D. Bailey (Referee)**

*1. The concept of "extensive" and "intensive" variables is still a bit confusing to me. I can think of examples such at top melting on sea ice (sidmassmelttop) that I could argue either way. I would request that in the appendices C-F the authors add the label of extensive or intensive to the variables, so that we in the community can be on the same page.*
**Change made:** We have added as appendix H a table listing all variables grouped by priority and intensive/extensive. We believe that this overview will be helpful for any modelling group in preparing its output for CMIP6 simulations

*2. The concept of sea ice freeboard is interesting and challenging even for the remote sensing community. I believe more description should be added here so as to make it clear what satellite product(s) we would be comparing the models against.*
**Changes made:** We have added: "This follows the classical definition of freeboard for in-situ observations. In the satellite community, sometimes the total height of sea ice and snow above sea level is referred to as freeboard. This can easily be calculated by adding sisnthick to sifb."

*3. Another issue is that most models that I am aware of do not keep track of terms for the sea ice and snow on top of sea ice separately. I would recommend that the following be expanded/clarified as to what is preferred: sidmassevapsubl and sndmassdyn. I know that in the CICE model we do not separate these for sea ice/snow.*
**Changes made:** Regarding *sidmassevapsubl:* We have clarified this by writing for *sidmassevapsubl* "If a model does not differentiate between the sublimation of snow and sea ice, we recommend to report sidmassevapsubl as zero as long as the ice is snow covered, and to report any sublimation within sndmasssubl." and for *sndmasssubl* "If a model does not differentiate between the sublimation of snow and sea ice, we recommend to report all sublimation within sndmasssubl as long as the ice is snow covered."

**Comments by Referee 2**

*-Appendix E. p.20. l.3: I find it confusing to put "downward always positive" in the title here and have fluxes defined positive upward l.15, l.20 and in the following page. I am fine with this convention but I suggest*

*phrasing it differently in the title l.3 e.g. "usually positive downward except when the term upward appears in the name description"*
**Changes made:** We share this concern and have simply removed the sign convention from the title of this appendix. We hence only clarify the sign convention in detail within the actual text of the appendix ("The sign convention is generally positive downward. However, to remain consistent with the NetCDF Climate and Forecast (CF) convention, upward fluxes that carry the term ``upward'' in their name are positive upward, as detailed in the following.")

*-Appendix F p.22 l.4: To me the request of sea ice speed is redundant with the x and y components of sea ice velocity. Why can't we estimate sea ice speed only from siu and siv ? In addition, I don't understand what is meant by " to account for back-and-forth movement of sea ice". Isn't the speed an absolute value that is positive by definition and hence does not provide any information about the direction? Please clarify the explanation and remove sispeed from the requested variables if it can be estimated from siu and siv.*
**Changes made:** We have clarified the definition to now read "Speed of ice (i.e. mean absolute velocity) to account for back-and-forth movement of the ice during the average period. Such change of direction during the averageing period may reduce the individual vector quantities siu and siv, which makes the calculation of the true sea-ice speed impossible. We hence ask for the absolute sea-ice speed separately. Requested as daily and monthly average."

*-p.23, l.10: P, h, C and A are not defined. I guess that sea-ice experts who use a Hibler model would understand but it would be useful to add at least a reference where this term is clearly defined.*
**Changes made:** We have now added the definition of the individual terms: "For Hibler-type models, this is the "ice pressure" $P (= P * \cdot h \cdot \exp(-C(1 - A)))$, where h is ice thickness, A is concentration and both $P *$ and C are empiricial constants."

*Typos:*
*-p.10, l.12, I guess the authors meant to use the word "even" instead of "ever"*
**Reply:** We indeed mean "ever".

*-p.17, l.15: "refozen" should be replaced by "refrozen"*
*-p.18, l.16 : the parenthesis should be closed after "advection..."*
**Changes made:** All corrected, thanks for spotting them

**Comments from CMIP Panel**

*The CMIP Panel is undertaking a review of the CMIP6 GMD special issue papers to ensure a level of consistency among the invited contributions, also in answering the key questions that were outlined in our request to submit a paper to all co-chairs of CMIP6-Endorsed MIPs. We very much welcome the important contribution from SIMIP to the CMIP6 special issue, below are a few comments:*

*Please ensure that the title of your paper includes both the acronym of the MIP, and CMIP6, so that it is clear this is a CMIP6-Endorsed MIP.*
**Changes made:** The title now reads: "The CMIP6 Sea Ice Model Intercomparison Project (SIMIP): Understanding sea ice through climate-model simulations"

*Please consistently use the term 'CMIP6-Endorsed MIPs' when you refer to other MIPs that are endorsed by CMIP6 (e.g. p2, l21; p10, l25)*
**Changes made:** We now refer to all MIPs as "CMIP6-endorsed"

*Please ensure consistency of the experiment names and abbreviations with the CMIP6 overview paper (Eyring et al., 2016): for example p5, l10: replace 'control simulation' with 'pre-industrial control simulation'.*
**Changes made:** This has been corrected

*p5, l10: while the piControl and the CMIP6 historical simulations are specified as experiments from which the model output defined in SIMIP is requested, there is no mentioning of the Diagnostic, Evaluation and*

*Characterization of Klima (DECK) experiments. Please specify whether the output should be collected also from the other CMIP DECK experiments (i.e. amip, abrupt-4xCO2 and 1pctCO2) and if so, why.*
*p5, l 10: is it necessary to collect all output from all CMIP6-Endorsed MIP experiments or can this be partly reduced to priority 1 output for example? Please could you be specific in the paper?*
**Changes made:** We now specify the additional experimental requests as follows: "We also request output from the CMIP6 DECK experiments abrupt-4xCO2 and 1pctCO2 as described by Eyring et al. (2015), as these allow direct insights into the equilibrium and transient sensitivity of sea ice to changes in the external forcing. For these experiments, which we will analyse as part of SIMIP, we request full storage of SIMIP variables. As detailed below, we recommend storage of priority 1 variables for other CMIP6-endorsed experiments independent of our data request, as the priority 1 variables will allow for a basic analysis of sea-ice evolution from any large-scale climate simulation."

*p2, l28: please replace 'CMIP6 data call' with 'CMIP6 data request' and you could refer to the invited contribution to this special issue or the website of the CMIP6 data request.*
**Changes made:** We have changed the wording and now refer to the website of the data request.

*p10, l15ff: any plans to contribute or encourage the contribution of observations that could be used to evaluate the proposed experiments to obs4MIPs?*
**Reply:** We will have two dedicated workshops later this year and early next year to specify the evaluation protocol. We expect that during these workshops, additional observational datasets will be identified that will then be contributed to obs4MIPs.

*Appendix A: Model documentation request (p.12): detailed model documentation including information on tuning is clearly important. However, this information should be collected as part of the Earth System Documentation (ES-DOC) activity (see http://es-doc.org) rather than in a separate effort. Please ensure the information that SIMIP requires is communicated to the ES-DOC group.*
**Changes made:** We now refer to the ES-DOC activity and their website for the documentation request.

*Appendix B (p 13, l8ff). We agree it is best to collect all variables on the native model grids. However, some additional information from the models is required to allow re-gridding of the data to a common grid. OMIP is proposing a weights file that model groups should provide to enable regridding from the native grid to one or two CMIP6 standard grids. Please refer to Griffies et al. (2016) and follow the same procedure for sea ice requests.*
**Changes made:** We now indicate that grid areas should be provided: "Note that analogues to the CMIP6-endorsed OMIP (Griffies et al., 2016), we request that files containing cell areas for the oceanic (areacello) and atmospheric (areacella) grid are supplied as part of the SIMIP output to allow for the correct weighting of individual grid cells."

*Appendices C-G: This is a very helpful overview of the variables requested by SIMIP. It would be nice to identify for each variable whether this is a variable that can (at least in principle) be evaluated with observations. Are simulators such as the COSP simulator required for any model-observation comparisons?*
**Reply:** We fully agree that this is helpful and indeed necessary in due course. This will hence be a central discussion item of the two upcoming workshops. The results, and links to the respective data sets, will then be published on the SIMIP website.

[revised manuscript text omitted]

---

## Author Response (AR2)

**The CMIP6 Sea Ice Model Intercomparison Project (SIMIP): Understanding sea ice through climate-model simulations**

Dirk Notz[1], Alexandra Jahn[2], Marika Holland[3], Elizabeth Hunke[4], François Massonnet[5,6], Julienne Stroeve[7,8], Bruno Tremblay[9], and Martin Vancoppenolle[10]

[1]Max Planck Institute for Meteorology, Hamburg, Germany
[2]Department of Atmospheric and Oceanic Sciences and Institute of Arctic and Alpine Research, University of Colorado at Boulder, Boulder, USA
[3]Climate and Global Dynamics Laboratory, National Center for Atmospheric Research, Boulder, USA
[4]Theoretical Division, Los Alamos National Laboratory, Los Alamos, New Mexico, USA
[5]Earth Sciences Department, Barcelona Supercomputing Center (BSC-CNS), Barcelona, Spain
[6]Georges Lemaître Centre for Earth and Climate Research, Earth and Life Institute, Université catholique de Louvain, Louvain-la-Neuve, Belgium
[7]National Snow and Ice Data Center, Boulder, USA
[8]University College London, London, UK
[9]Department of Atmospheric and Oceanic Sciences, McGill University, Montreal, Canada
[10]Sorbonne Universités, UPMC Paris 6, LOCEAN-IPSL, CNRS/IRD/MNHN, France

*Correspondence to:* D. Notz (dirk.notz@mpimet.mpg.de)

Dear Dr. Valcke,

thank you very much for your careful reading of the revised manuscript and for your helpful comments. We apologize for not including a version with track-changes, and have now included such version below. It includes all changes relative to the version that was put online for GMDD.

5    Regarding your major comment on grid information, we now follow the paper by Griffies et al. (2016) by specifically including the request that all groups should provide sufficient information to allow for the re-gridding of their output. As Grieffies et al. (2016), we refer to the upcoming paper by the WGCM Infrastructure Panel for further details.

We have made all minor changes that you pointed out.

In addition, we've been made aware of two small inconsistences of our paper with the official data request. These were

10    caused by some last minute changes in the data request that did not make it into the paper. They refer to the fact that in the official data request we ask for melt-pond mass rather than for melt-pond area, and we ask for the output of drainage of freshwater from the sea-ice surface. To overcome these inconsistencies, we have corrected these two issues in the attached version of our manuscript.

Please get in touch if any further issues should come up.

15    Thank you once again for your help and support,

with best regards,

Dirk Notz (for all authors)

[revised manuscript text omitted]

<table>
<tr><td colspan="3" align="center">Priority 1, intensive</td></tr>
<tr><td>Sea-ice thickness</td><td>sithick)</td><td>mon, day</td></tr>
<tr><td>Snow thickness</td><td>sisnthick</td><td>mon, day</td></tr>
<tr><td>Surface temperature</td><td>sitemptop</td><td>mon, day</td></tr>
<tr><td>X-component of sea-ice velocity</td><td>siu</td><td>mon, day</td></tr>
<tr><td>Y-component of sea-ice velocity</td><td>siv</td><td>mon, day</td></tr>
<tr><td>Sea-ice speed</td><td>sispeed</td><td>mon, day</td></tr>
<tr><td>beginSQUAREBRACKET3ex]</td><td colspan="2" align="center">Priority 1, extensive</td></tr>
<tr><td>Sea-ice area fraction</td><td>siconc</td><td>mon, day</td></tr>
<tr><td>Sea-ice mass per area</td><td>simass</td><td>mon</td></tr>
<tr><td>Snow area fraction</td><td>sisnconc</td><td>mon</td></tr>
<tr><td>Snow mass per area</td><td>sisnmass</td><td>mon</td></tr>
<tr><td>Sea-ice volume per area</td><td>sivol</td><td>mon</td></tr>
<tr><td>beginSQUAREBRACKET3ex]</td><td colspan="2" align="center">Priority 1, temporal average of 1-D time series</td></tr>
<tr><td>Fraction of time steps with sea ice</td><td>sitimefrac</td><td>mon</td></tr>
<tr><td>Sea-ice area North</td><td>siarean</td><td>mon</td></tr>
<tr><td>Sea-ice area South</td><td>siareas</td><td>mon</td></tr>
<tr><td>Sea-ice volume North</td><td>sivoln</td><td>mon</td></tr>
<tr><td>Sea-ice volume South</td><td>sivols</td><td>mon</td></tr>
<tr><td>Sea ice extent North</td><td>siextensiveentn</td><td>mon</td></tr>
<tr><td>Sea ice extent South</td><td>siextensiveents</td><td>mon</td></tr>
<tr><td>Sea-ice-mass flux through straits</td><td>simassacrossline</td><td>mon</td></tr>
<tr><td>beginSQUAREBRACKET3ex]</td><td colspan="2" align="center">Priority 2, intensive</td></tr>
<tr><td>Temperature at snow-ice interface</td><td>sitempsnic</td><td>mon</td></tr>
<tr><td>Temperature at ice-ocean interface</td><td>sitempbot</td><td>mon</td></tr>
<tr><td>Age of sea ice</td><td>siage</td><td>mon</td></tr>
<tr><td>Sea-ice or snow albedo</td><td>sialb</td><td>mon</td></tr>
<tr><td>Sea-ice freeboard</td><td>sifb</td><td>mon</td></tr>
<tr><td>Downwelling shortwave flux over sea ice</td><td>siflswdtop</td><td>mon</td></tr>
<tr><td>Upward shortwave flux over sea ice</td><td>siflswutop</td><td>mon</td></tr>
</table>

[revised manuscript text omitted]